# Pickering Emulsions: A Novel Tool for Cosmetic Formulators

**Eduardo Guzmán** [1,2,*], **Francisco Ortega** [1,2] **and Ramón G. Rubio** [1]

[1] Departamento de Química Física, Facultad de Ciencias Químicas, Universidad Complutense de Madrid, Ciudad Universitaria s/n, 28040 Madrid, Spain; fortega@quim.ucm.es (F.O.); rgrugio@quim.ucm.es (R.G.R.)

[2] Instituto Pluridisciplinar, Universidad Complutense de Madrid, Paseo Juan XXIII 1, 28040 Madrid, Spain

[*] Correspondence: eduardogs@quim.ucm.es; Tel.: +34-913-944107

**Abstract:** The manufacturing of stable emulsion is a very important challenge for the cosmetic industry, which has motivated intense research activity for replacing conventional molecular stabilizers with colloidal particles. These allow minimizing the hazards and risks associated with the use of conventional molecular stabilizers, providing enhanced stability to the obtained dispersions. Therefore, particle-stabilized emulsions (Pickering emulsions) present many advantages with respect to conventional ones, and hence, their commercialization may open new avenues for cosmetic formulators. This makes further efforts to optimize the fabrication procedures of Pickering emulsions, as well as the development of their applicability in the fabrication of different cosmetic formulations, necessary. This review tries to provide an updated perspective that can help the cosmetic industry in the exploitation of Pickering emulsions as a tool for designing new cosmetic products, especially creams for topical applications.

**Keywords:** coalescence; colloidal particles; creams; dispersions; emulsions; stability

## 1. Introduction

Emulsions are colloidal dispersions formed by two immiscible liquids, where one of them is dispersed as droplets (dispersed phase) into the second one (continuous phase). However, the high surface energy of the interface between the two immiscible liquids results in thermodynamic instability. This leads to emulsion destabilization through different processes, e.g., flocculation, creaming, coagulation, coalescence, phase inversion, and Ostwald ripening, although they can be kinetically arrested for long periods of time [1]. The most common approach exploited for stabilizing emulsions is the use of surfactants or amphiphilic polymers, which contribute to the reduction in the interfacial tension between the two fluids by forming a molecular layer around the liquid droplets. This enhances the kinetic stability of emulsions, minimizing the destabilization events. However, it must be considered that the preparation of kinetically stable emulsions is a very difficult task due to the complexity of their interfacial and rheological behavior [2,3]. Despite the multiple difficulties associated with emulsion preparation, these systems are versatile tools for the cosmetic industry, finding extensive exploitation in the production of cosmetics for different purposes, e.g., sebum control, encapsulation, color cosmetics, skin whitening or UV protection [4–6].

Cosmetic emulsions are commonly very complex polydisperse multicomponent mixtures, consisting on the combination of the two liquids (water/hydrophilic base liquid and oil/hydrophobic base one), more than one active surface molecule (stabilizers), and several additives. The latter can contribute to improving the functionality or the sensorial feeling associated with the use of the formulation, provide fragrance, or simply enhance the quality of the final product (e.g., increase their stability or modify their viscosity and texture) [6,7]. The most common stabilizers of cosmetic emulsions are conventional surfactants, e.g., tween 80, span 80, sodium laureth sulfate (SLES), sodium lauryl sulfate (SLS), sodium dodecyl sulfate (SDS), cocamidopropyl betaine (CAPB) or polyethyleneglycol ethers [8–10].

However, in most cases, molecular and polymeric surfactants are not enough to guarantee the long-term stability of cosmetic emulsions and prevent a change in their properties over time [4]. Moreover, the use of synthetic surfactants in cosmetic products may cause adverse effects, e.g., allergies, hemolysis or cytotoxicity, to the final consumer and limit the eco-sustainability profile of cosmetics. The most common emulsifiers used in the cosmetic industry may induce different adverse effects depending on their chemical nature. For instance, cationic emulsifiers are more toxic than anionic ones, and the latter are more toxic than the non-ionic ones. Despite this toxicity, emulsifiers are widely used in cosmetic products, with anionic surfactants having a prominent role as emulsifying agents. In fact, the low cost and stability of anionic surfactants have expanded their use in a broad range of cosmetic products; unfortunately, they may cause skin irritation due to their ability to modify the multilamellar structure lipid structure of the stratum corneum [11,12]. On the other side, the recovery of surfactant residues is not always easy from a practical point of view [13]. Therefore, it is urgent to seek alternative species, allowing the stabilization of emulsions to manufacture safer and more eco-sustainable cosmetic emulsions with long-term stability and satisfactory consumer perception during their application. A very promising approach for improving the quality of cosmetic emulsions is the replacement of conventional surfactant with colloidal particles to obtain particle-stabilized emulsions, the so-called Pickering emulsions [14]. This type of emulsion has become a very interesting alternative for the cosmetic industry. However, to date, there are many difficulties in manufacturing stable emulsions that provide a suitable sensorial perception upon application. Figure 1 schematizes the main features of emulsions stabilized by surfactants and particles.

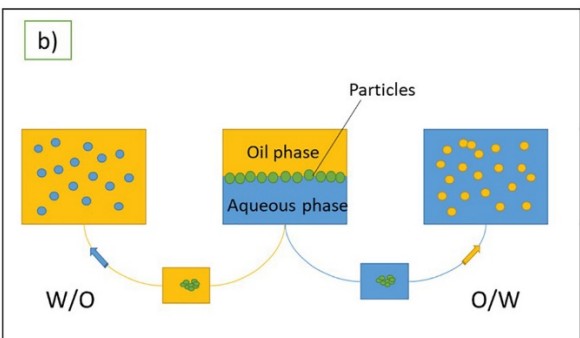

**Figure 1.** Sketch of the main features of surfactant-stabilized (**a**) and particle-stabilized (**b**) emulsions. Adapted from Venkataramani et al. [7], Copyright (2020), with permission from Elsevier.

Pickering emulsions were first described by Ramsdem and Pickering more than one century ago [15,16] and have undergone a very important development in the last two decades. Nowadays, it is possible to fabricate Pickering emulsions with properties selected at will, which opens up many opportunities for the design of new products in different industries, including the cosmetics industry [7,17]. In fact, the use of particles allows controlling the nature of the emulsions, the organization of the droplets and viscosity, modifying the sensory feeling upon application, product appearance and texture [18]. This makes possible the fabrication of innovative cosmetic formulations and multifunctional delivery systems. Moreover, Pickering emulsions offer many interesting properties compared to conventional ones, including superior stability against coalescence, minimization of the Ostwald ripening phenomena, higher biocompatibility and low cytotoxicity [19].

This review tries to present a general overview of the potential interest of Pickering emulsions in cosmetics, highlighting some of the areas where they can contribute most to the current aims of the cosmetics industry. It is true that to date, the number of studies dealing with the use of particles for the stabilization of cosmetic emulsions remains scarce.

However, Pickering emulsions can help to open up new avenues for the optimization of new eco-sustainable cosmetic products with enhanced sensorial perception.

## 2. General Aspects of Pickering Emulsions

### *2.1. Preparation of Pickering Emulsions*

Pickering emulsions can be obtained using any methodology commonly used for the preparation of surfactant-stabilized emulsions. However, when Pickering emulsions are considered, it is preferable to use high-energy methods, such as rotor-stator or high-pressure homogenization, or sonication, to ensure that the formation of droplets can occur on a similar time scale as the trapping of particles to the nascent droplet/continuous phase interface. In recent years, the use of membrane emulsification and microfluidic techniques has received growing interest in the preparation of Pickering emulsions [20].

It should be noted that among the many techniques available for the preparation of Pickering emulsions, their scaling up is only possible by using rotor-stator and high-pressure homogenization processes. However, the development of other methodologies remains very interesting for improving product formulation. In particular, the use of low shear rate methodologies, such as membrane or microfluidic emulsification, may be very interesting for the cosmetics industry, allowing the fabrication of emulsions with reduced polydispersity. Furthermore, this type of method does not induce any significant heating during the preparation process, which can be very interesting for the preparation of emulsions involving thermo-labile compounds.

The emulsification process is one of the most important aspects that can be modified for controlling emulsion properties, e.g., emulsion type or droplet size. However, there are many other parameters that can be exploited for modulating the emulsion properties. In the following, some details about the most common methods used for obtaining Pickering emulsions will be discussed.

### 2.1.1. Rotor-Stator Homogenization

The rotor-stator method relies on the production of droplets as the result of the high-speed rotation of a bladed rotor attached to a stationary stator. Thus, the formation of droplets is possible by the combination of the high-speed rotation of the liquid and the shear forces emerging between the rotor and the stator, allowing control of the droplet sizes by modifying the homogenization time and rotation speed [21].

This methodology offers many opportunities in the fabrication of emulsions due to their simplicity and low cost. Furthermore, the production of emulsions by using the rotor-stator homogenization is relatively fast and flexible. However, the poor uniformity of the obtained droplets emerges as a very important drawback of this type of methodology [20].

### 2.1.2. High-Pressure Homogenization

High-pressure homogenization is a multistep process that can be classified as a continuous emulsification process. This involves an initial pre-emulsification, leading to the formation of coarse emulsions, which are later passed through the slits of the high-pressure homogenizer, where a fine emulsion is obtained by the combination of cavitation, turbulence and shear forces to prepare the primary coarse emulsion into a fine emulsion [22]. It is common that the high-pressure homogenization process leads to the formation of more homogeneous droplets with a smaller size than when rotor-stator homogenization is used. However, the high shear rate deformations can alter the physical integrity of soft colloidal particles and agglomerates [20].

### 2.1.3. Ultrasound-Assisted Emulsification

The use of ultrasounds for the preparation of emulsification relies on the application of radiation of a frequency above 16 kHz, which can interact with substances to obtain emulsions [20]. This is possible due to cavitation phenomena resulting from the interaction between the mixture and the ultrasound radiation. Thus, cavitation causes a local increase

in the temperature and pressure, which leads to turbulence flow and shear stresses, favoring the adsorption of the emulsifiers on the droplet surface and the stabilization of Pickering emulsions with similar properties to those obtained by high-pressure homogenization. Unfortunately, the energy consumption of ultrasound-assisted emulsification is high, which results in high operational costs [20].

### 2.1.4. Membrane Emulsification

Membrane emulsification relies on the preparation of emulsions by pressing a pure dispersed phase or primary coarse emulsions through a microporous membrane with a controlled injection rate and shearing conditions. The properties of the obtained emulsions are dependent on the size of the pore of the membrane, the viscosity of liquid phases and the interfacial tension. Membrane emulsification is a low-energy method that allows for obtaining emulsions with uniform droplets. However, it is time-consuming and can only be used for systems with low viscosity [23].

### 2.1.5. Microfluidic Technologies

The use of microfluidic technologies for manufacturing emulsions relies on the formation of droplets as the result of the drag force experienced by the dispersed phase when it meets the continuous one. This is possible by using a device that enables the flow of the continuous and dispersed phases perpendicular to each until reaching the contact point [24].

### 2.2. Understanding the Stabilization of Emulsions by Colloidal Particles

The stabilization of emulsions requires overcoming the thermodynamic instability arising from the high interfacial tension between the two immiscible liquids and the high free energy of the system. This is only possible by a correct choice of the emulsification conditions, including the stabilizer concentration and the mixing conditions (mixing time and speed) [25]. In the particular case of Pickering emulsions, the stabilization occurs as a result of the adsorption of colloidal particles, which can present a broad range of morphologies (fibrils, spheres, platelets, nanosheets, rods, cylinders or cubes) [26,27] to the liquid/liquid interface. However, in contrast to what happens when a molecular surfactant is considered, the adsorption of particles to the interface is not only associated with a reduction in the interfacial tension between the two liquids but also the subsequent minimization of the free energy of the system, and the formation of a particle layer leads to a physical barrier that obstructs contact between contiguous droplets. This increases the resistance against droplet coalescence, and hence, an enhancement of the emulsion stability in relation to a conventional one stabilized by the presence of molecular species may be expected. Therefore, the properties of the particles, mainly their wettability and size, as well as their concentration, can play a central role in the control of the emulsion stability [28]. Furthermore, the nature of the two liquids separated by the interface plays a very important role in the stabilization process of Pickering emulsions. In fact, the adsorption of particles to a fluid interface, and in particular, the position of the particles in relation to the interfacial plane, defined by the particle wettability, is the result of a mechanical equilibrium involving three interfacial tensions (two solid/liquid interfacial tensions and one liquid/liquid interfacial tension) [5,20]. This can be expressed in terms of Young's equation for perfectly spherical particles

$$cos\theta = \frac{\gamma_{po} - \gamma_{pw}}{\gamma_{ow}}, \tag{1}$$

where $\gamma_{po}$ and $\gamma_{pw}$ are the two solid/liquid tensions (particle/oil and particle/water interfacial tensions, respectively), and $\gamma_{ow}$ is the interfacial tension between the two liquids separated for the interface. $\theta$ is the three-phase contact angle corresponding to the particle trapped at the fluid interface and can be geometrically defined, as shown in Figure 2. According to Young's equation, the contact angle of a specific type of particle depends on the nature of the two liquids separated for the interface and determines what liquids are

in the continuous and dispersed phases, defining the emulsion type. Thus, hydrophilic particles characterized by $\theta < 90°$ drive the formation of oil-in-water (O/W) emulsions, whereas water-in-oil (W/O) emulsions are formed when hydrophobic particles with $\theta > 90°$ are used (see Figure 2). From a practical point of view, O/W emulsions are formed at $15° < \theta < 90°$, whereas the stabilization of W/O requires $90° < \theta < 165°$. Therefore, partial wetting for both phases is essential for stabilizing Pickering emulsions, and hence, particles fully wetted by one of the phases do not allow the stabilization of particle-stabilized emulsions [26,29]. It should be noted that in recent years, the research on multiple emulsions (oil-in-water-in-oil or water-in-oil-in water) or oil-in-oil emulsions has undergone a strong development [30,31].

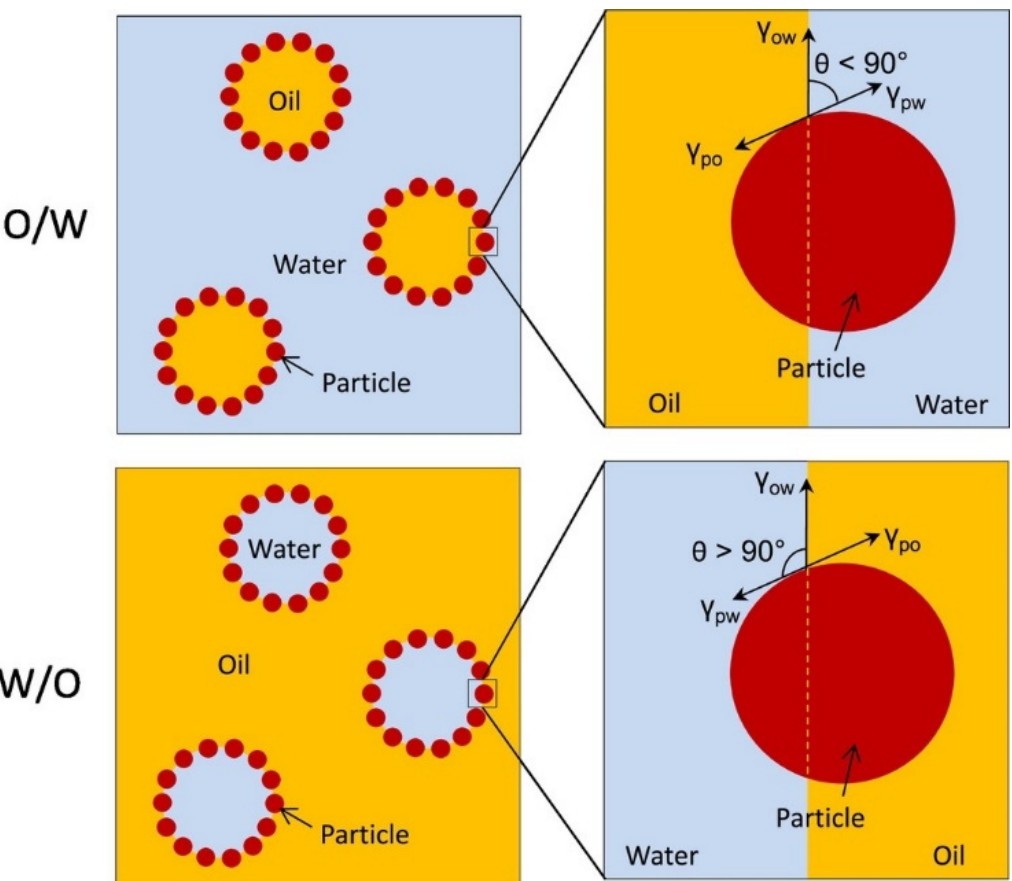

**Figure 2.** Sketch of the organization of the liquids and the particles in O/W and W/O Pickering emulsions (**left**) together with the geometrical definition of the three-phase contact angle ($\theta$) for both types of emulsions as a function of the particle–oil ($\gamma_{po}$), particle–water ($\gamma_{pw}$) and oil–water ($\gamma_{ow}$) interfacial tensions (**right**). Reprinted from Albert et al. [20], Copyright (2019), with permission from Elsevier.

The emulsion type is controlled by the specific interactions between the phases and the particles that is accounted for the contact angle, but the volume fraction of each phase is also a very important variable for tuning the emulsion type. For instance, if the volume fraction of the dispersed phase is close to 0.7, a phase inversion with respect to that expected for the contact angle value may occur. Thus, by tuning the volume fractions of the two liquids, it is possible to switch emulsions stabilized with hydrophilic particles from O/W emulsions to W/O one. A similar transition can occur for hydrophobic particles (in this case, the transition occurs from W/O emulsions to O/W emulsions) [32].

The importance of the particle wettability in Pickering emulsions goes beyond the control of the nature of the formed emulsion; it also affects the energy required to trap a particle to the interface between the two liquids, $\Delta E_p$, which can be defined as

$$\Delta E_p = -\pi R^2 \gamma_{ow} (1 \pm cos\theta)^2, \tag{2}$$

where $R$ defines the particle radius. The $\pm$ sign accounts for the different affinities of the particles for the fluid phases. Thus, for particles preferentially immersed in the upper phase (commonly the less polar one), the $+$ sign applies, whereas the sign $-$ applies for particles with preferential immersion in the bottom phase (generally the most polar one). In most the cases, except for very small particles (radius below 10 nm), the energy for trapping a particle to the fluid interface exceeds many times the thermal energy, $k_BT$, where $k_B$ is the Boltzmann constant and $T$ is the absolute temperature. Hence, a dynamic exchange of the stabilizing molecules attached to the interface cannot be expected as in conventional emulsions. Therefore, this irreversibility of the attachment of particles to the interface ensures the formation of a quasi-rigid armor on the droplet-continuous phase interface, which behaves like a physical barrier blocking the droplet contact as a result of volume exclusion phenomena, minimizing the coalescence and Ostwald ripening phenomena. This enhances the stability of particle-stabilized emulsions in relation to conventional ones [33]. It should be noted that the size distribution, i.e., the polydispersity of the particles used for stabilizing emulsions, may also influence the final properties of the obtained emulsions. This may be easily understood considering the subtle differences in the surface properties of particles of different sizes, which can affect their attachment to the droplet surface [34].

Particle concentration is also a very important parameter governing the properties of particle-stabilized emulsions. For instance, the higher the particle concentration, the smaller the dimension of the dispersed phase droplets. This can be understood considering that the increase in the number of particles allows for stabilizing a larger area, partially hindering the droplet coalescence. Furthermore, coalescence can also be prevented by forming a tightly packed particle layer on the droplet/continuous phase interface. On the other hand, the increase in the particle concentration beyond the maximum value required for covering all the interfacial area introduces a steric hindrance contribution, which favors the stabilization process. However, if the particle concentration is too high, aggregation and sedimentation phenomena can appear, fostering emulsion destabilization [35]. Moreover, the particle size also impacts the emulsion stability and the size of the dispersed droplets. In fact, it is common that the increase in particle size can be accompanied by an increase in droplet size. This may be rationalized considering the impact of the particle size on the trapping energy, which, as discussed above, controls the emulsion stability. On the other hand, big particles adsorb to the interface slower than small ones, which results in the formation of layers characterized by a weaker packing, and hence, their role in coalescence prevention is very limited [20].

There are many other aspects that may affect the stability of Pickering emulsions, e.g., particle shape, roughness and charge, and environmental conditions (pH or ionic strength). For instance, particle shape can modify the stabilization mechanism due to the different particle arrangements that can be formed at the interface as a result of the differences in the aspect ratio of the particles. It is common that the increase in the particle aspect ratio, i.e., asymmetry, can contribute to the stabilization of Pickering emulsions [20]. Moreover, other structural parameters of the particles, such as their flexibility or conformation, and the particle-laden interface can modify the interfacial coverage and, hence, the emulsion stability [26,36]. On the other hand, particle roughness may also modify the stability of emulsions due to their impact on the particle contact angle. In fact, particle roughness can induce pinning–depinning phenomena of the three-phase contact line, which, in turn, alters the trapping of the particles to the fluid interface [34].

The attachment of particles to the fluid interface and, hence, the stability of Pickering emulsions can also be modified by the particle charge. In fact, particles with reduced charge density provide a better stabilization of Pickering emulsions, whereas the increase in the

particle charge density introduces a repulsive contribution between the particles and particle-decorated droplets, which makes it very difficult to attain a very high particle coverage of the droplet/continuous phase interface, resulting in an increase in the emulsion instability [37].

The role of the environmental conditions, e.g., the ionic strength or pH of the aqueous phase, may alter the stabilization process of Pickering emulsions as a result of their impact on very important aspects such as the particle contact angle or charge. In particular, the change of the ionic strength or pH can modify the character of the interactions existing within the system, which in turn affect the emulsion stability [20]. Furthermore, the ionic strength and pH can alter the effective hydrophilic–lipophilic balance of particles, modifying their ability to remain trapped at fluid interfaces [26].

*2.3. Rheology of Pickering Emulsions*

Rheology is a scientific branch dealing with the response of materials under deformation and flow conditions [38]. The rheological properties are of paramount importance to the control of the stability of emulsions, which is essential for the successful application of this type of system [39,40]. In fact, the rheological characterization of Pickering emulsions can provide very important information related to the flow characteristics of the emulsions as well as the emulsion stability [28].

The rheological performance of surfactant-stabilized and particle-stabilized emulsions are very different. Mason et al. [41] demonstrated that the elastic modulus of surfactant-stabilized emulsions depends on the volume fraction of the oil phase, increasing drastically when the volume fraction of the dispersed phase overcomes the threshold value of 0.63. This is ascribed to a random close packing of spherical droplets. On the contrary, the elastic modulus of Pickering emulsions depends on the interfacial elasticity as a result of the strong adhesion of the particles to the interface, presenting a rheological response mainly governed by the volume fraction of the particles. In fact, it is possible to find very a similar rheological response in Pickering emulsions with the same particle ratio independently of the volume fractions of the liquid phases [42,43]. Thus, Pickering emulsions present elastic-like behavior even for volume fractions of the dispersed phase that are well below the random-close-packing-limit of spheres. This enhances their mechanical stability in relation to surfactant-stabilized emulsions, hindering the phase separation events [7]. It should be noted that despite the fact that most Pickering emulsions exhibit gel-like behavior, it is possible to modulate their rheological response by changing different parameters, including the particle concentration or the volume fraction of the oil phase [35]. This is very important for optimizing the application of cosmetics formulation. Thus, the manufacturing of emulsions with shear thinning behavior and a mainly elastic response (storage component higher than the loss one) facilitates the application of the emulsions within cosmetic substrates [44].

The increase in the particle concentration is associated with a reduction in the emulsion viscosity, which leads to a worsening of the emulsion stability. This is in agreement with the stabilization of the interface mediated by the steric hindrance associated with the particle adsorption to the fluid interface [17,45]. On the other hand, a very important aspect of the rheological behavior of Pickering emulsions for their practical use is their thickening character. This can be understood considering that the high viscosity of the emulsion can slow down the destabilization events. In fact, thickening plays a very important role in emulsion stabilization [46]. Moreover, emulsion thickening is very important in cosmetic emulsions to ensure a smooth sensorial feeling upon their application. Therefore, the viscosity of cosmetic emulsions should be adjusted in such a way that the emergence of extreme rheological behaviors (too runny or too viscous emulsions) can be avoided, but these emulsions should remain stable for a long time under storage conditions in their containers [7].

The ability of particles to stabilize Pickering emulsions is commonly associated with their ability to adsorb to the droplet/continuous phase interface, forming highly packed networks around the droplet, which confer rigidity and viscosity to the interface [47,48].

This explains the similar rheological behavior found for Pickering emulsions obtained with oil phases of different polarities [45].

### 2.4. Destabilization Mechanisms of Emulsions

It is common that the emulsion properties can remain unchanged for a certain period of time. This defines the so-called emulsion stability, even though it cannot be defined as a true thermodynamic characteristic, and hence, emulsion properties undergo changes over time. Therefore, the velocity of such changes determines the emulsion stability. Thus, according to the definition of the interfacial free energy ($\Delta G = \gamma_{ow}\Delta A$, with $\Delta A$ being the change of the total interfacial area of the system), it may be expected that the increase in the interfacial area upon emulsification can lead to the destabilization of the system, and hence, this unfavorable contribution must be counteracted by the negative change of energy associated with the trapping of the particles to the interface. This is an irreversible process that drives a kinetic arrest of the system, i.e., Pickering emulsions present kinetic stability [49–52].

Pickering emulsion destabilization is accompanied by the separation of the macroscopic phases and can occur following different mechanisms, which can occur individually or coupled. Figure 3 schematizes the different destabilization mechanisms of emulsions that can drive phase separation.

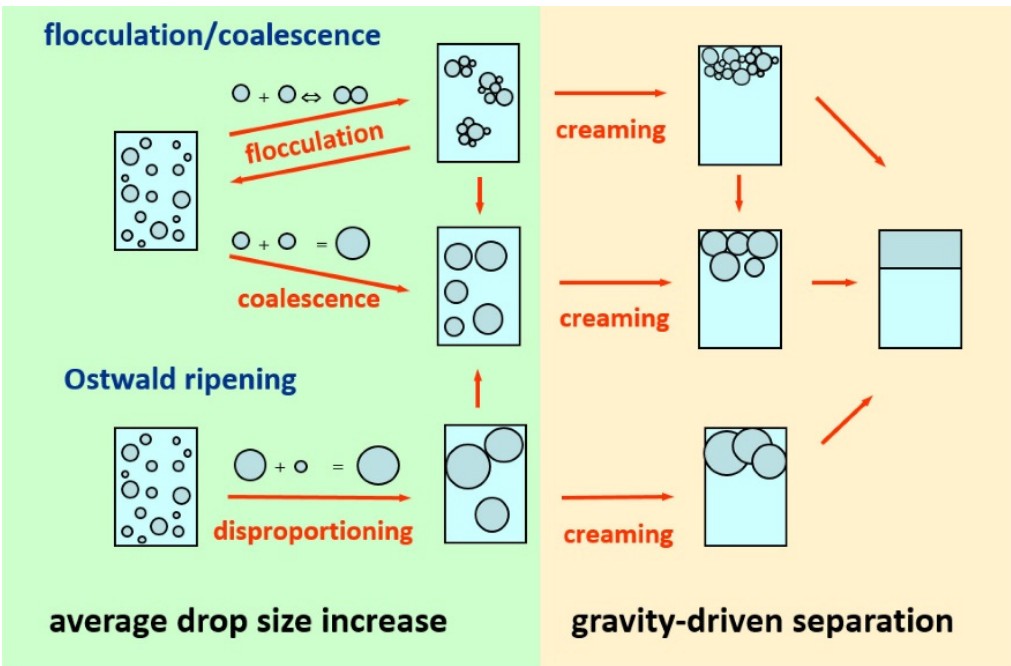

**Figure 3.** Sketch of some destabilization mechanisms that can drive the phase separation in emulsions. Notice that sedimentation and creaming are similar processes and do not occur simultaneously, depending on the emulsion nature. Therefore, for the sake of example, only the creaming is shown in the sketch. Reprinted from Ravera et al. [3], Copyright (2021), with permission from Elsevier.

Droplets with higher density than the continuous phases can settle down and form a separated liquid layer at the bottom of the emulsion. On the contrary, when the droplets present a lower density than the density of the continuous phase, they tend to rise up, which results in the formation of a liquid layer on top of the emulsion. These two destabilization processes are called sedimentation and creaming, respectively, and can be classified as gravity-driven separation, with their occurrence being strongly dependent on the nature of the dispersed phase [3,47]. On the other hand, the association of several droplets to form larger aggregates without a true coalescence can also contribute to the emulsion destabilization through flocculation processes. Ostwald ripening is especially important

in polydisperse emulsions, and it results from the growth of the smallest droplets due to the mass diffusion from the disperse phase through to the continuous phase toward the larger droplets due to the difference in Kelvin pressure. The last destabilization event is the coalescence, which is characterized by a fusion of individual droplets into a larger one as a result of the thinning and disruption of the fluid film between the droplets [26]. It is common that emulsion destabilization proceeds following a two-step process. Firstly, the size of the droplets increases by flocculation, coalescence or Ostwald ripening, and then when the droplets reach a critical dimension, the gravitational forces start to operate, driving the formation of phase-separated liquid layers [3]. The minimization of the destabilization events is possible by a correct choice of the emulsifier substance, or the emulsion rheology, to produce emulsions with a narrow distribution of the size of the droplets [29]. It should be noted that together with the physical stability evaluated by the absence of phase separation and coarsening, the preparation of cosmetic emulsions should present chemical and microbiological stabilities [7].

## 3. Particles in Cosmetic Pickering Emulsions

Nano- and microparticles present excellent characteristics for their use in cosmetics, opening up very important avenues for the fabrication of new products with improved formulation, delivery of active ingredients, skin penetration or long-lasting effects [53]. For instance, the existence of particles with a broad range of sizes and surface chemistries can be an advantage of topical applications, contributing to controlling the skin penetration of the emulsions. Moreover, they can adsorb, forming occlusive layers, which can contribute to an increase in the retention time of active ingredients contained in the formulations [54]. In fact, the many possibilities offered by particles in cosmetics have extended their use to multiple families of cosmetic products, including facial makeup, nail, skin, and hair care products [7].

The choice of the most suitable particles for a specific application is essential due to their multiple roles, e.g., emulsion stabilization, film formation, moisturizing and exfoliation, conditioning and waterproofing properties [55]. Nowadays, there is a broad range of particles that can be exploited for the design of cosmetically acceptable formulations [19].

- Polymer particles of different nature (both natural and synthetic) are currently exploited mainly in skin care products, including anti-aging and moisturizing creams. For instance, chitosan particles, a copolymer of D-glucosamine and N-acetyl-D-glucosamine, are widely used due to their natural abundance, biocompatibility and biodegradability. On the other side, hyaluronic acid is also very common in the cosmetics industry due to its anti-wrinkle properties [56].
- Silver nanoparticles are very common in cosmetic products due to their antimicrobial and antifungal properties. In fact, silver nanoparticles are common in toothpastes, creams, soaps, lotions or deodorants [57].
- Gold nanoparticles are used to help with the delivery of different types of molecules through the skin. Their penetration into the stratum corneum shows a strong dependence on its physico-chemical properties, e.g., shape, size and chemical surface, and compatibility with the lipid domains existing within the skin. Moreover, the use of gold particles in cosmetics is gaining interest due to their antioxidant and antimicrobial properties, contributing to improvements in skin firmness and elasticity. This has pushed their incorporation into anti-aging creams, lotions and deodorants [58].
- Titanium oxide and zinc oxide nanoparticles are commonly incorporated in cosmetic products for providing UV filter properties due to their capacity for reflecting the UVB and UVA radiations, respectively. In fact, the combination of both oxides provides good protection against sun radiation, allowing to prepare transparent products with good spreadability and texture. Moreover, they reduce the skin irritation associated with most of the chemical UV filters. The mechanisms of these types of particles for providing UV protection are related to their ability to be deposited on the external surface of the stratum corneum [59].

- Silica nanoparticles, commonly in the diameter range of 5–100 nm, have received interest in the cosmetic industry mainly as a result of some of their specific characteristics, e.g., their pleasant sensorial properties and their capacity for delivery of lipophilic and lipophobic compounds. Furthermore, this type of material can be easily obtained in large amounts using low-cost processes and can be chemically modified to obtain specific characteristics. This has stimulated the use of silica nanoparticles in toothpastes, makeup products, hair styling, deodorants and skin care. On the other hand, silica nanoparticles can be used as emulsifiers, emollients and water barriers, having an adjuvant effect in sun protection products because they contribute to the enhancement of their spreadability, reducing the degradation of the products [60].

## 4. Pickering Emulsions in Cosmetics

The use of Pickering emulsions for cosmetic purposes can contribute to eliminating, or at least significantly reducing, the toxicity and irritation arising from the use of conventional surfactants. Simultaneously, the use of particles can contribute to enhancing the droplet rigidity, which can be a good choice for embedding, sustained release and transdermal adsorption of active ingredients [61]. However, the development of Pickering emulsions for cosmetic applications is limited due to the scarce number of available studies dealing with the use of cosmetically acceptable compounds. In fact, most of the studies about Pickering emulsions deal with the use of non-biocompatible oil phases, including toluene, n-dodecane or hexadecane, which limits the validity of the extracted conclusions because the oil nature can strongly modify the emulsion stability [17]. This was explored by Wu et al. [62] by preparing Pickering emulsions stabilized with silica nanoparticles (diameter of 160 nm) using different cosmetically acceptable oils (silicone oils and ester oils). They found that, depending on the oil nature, it was necessary to use particles of different nature for preparing O/W and W/O emulsions. The different penetration of emulsions stabilized with quinoa starch particles containing different oil phases (miglyol, paraffin and sheanut oil) was investigated by Marku et al. [63]. They found that the stabilization of cosmetically acceptable emulsions requires a minimal oil volume fraction of about 56%. On the other hand, the droplet size was only dependent on the weight ratio between the oil phase and the starch particles, which is very important for controlling the rheological and cosmetic properties of creams containing Pickering emulsions but did not play any role in the control of the transdermal diffusion of the formulation.

Terescenco et al. [18] compared the textures of Pickering emulsions stabilized with three different types of particles ($TiO_2$, $SiO_2$ and ZnO) with that of conventional emulsions stabilized by surfactants. They found that particle-stabilized emulsions result in a less glossy and greasy texture than surfactant-stabilized emulsions, enhancing the spreadability of the formulation. In fact, Pickering emulsions present a better sensorial profile than conventional emulsions, which is evidenced in their improved sensorial effects, absorbance, spreadability and stability [64]. Moreover, the specific chemistry of the particles is essential for controlling the sensory perception of the emulsions upon application [18].

The design of delivery systems is accounted for among the most important potential uses of Pickering emulsions in cosmetics. Wei et al. [65] designed O/W Pickering emulsions stabilized using poly(lactide-co-glycolide) (PLGA)/poly(styrene-co-4-styrene-sulfonate) (PSS) particles for the encapsulation within the inner core of an lipophilic active ingredient (tocopheryl acetate). This approach allowed for an encapsulation efficiency higher than 98% of the active molecule within spherical droplets of about 2 μm, which remain stable in the pH range compatible with their application for skin care purposes (4.3–7.1). Moreover, the obtained product presents good long-term stability, protecting the antioxidant activity of the tocopheryl acetate. On the other hand, the encapsulation process does not affect the cellular uptake of the active ingredient. However, it significantly enhances its activity. Pickering emulsions were also used as delivery platforms by Sharkawy et al. [66]. They fabricated Pickering emulsions stabilized with chitosan/gum arabic nanoparticles for topical delivery of trans-resveratrol. This type of emulsion contributes to enhanced delivery of the active

compound through the epidermis and dermis. The strong ability of the chitosan/gum arabic nanoparticles for emulsion stabilization exploits their quasi-neutral wettability (contact angle close to 90°), which ensures their long-term stability. This is especially evident for emulsions with high oil volume fractions (0.6–0.7), which present a shear-thinning response accompanied by an elastic-like behavior [67]. On the other hand, this type of formulation ensures better dermal delivery of the encapsulated molecules, with a five-fold increase in skin retention of the trans-resveratrol in comparison with their solutions in ethanol. This is ascribed to the cationic nature of the chitosan, which contributes to the opening of the tight junctions of the negatively charged stratum corneum cells, facilitating the accumulation of the molecule within the skin. Moreover, the encapsulation within emulsions of trans-resveratrol ensures their photo-stability [66].

Arriagada et al. [68] explored the simultaneous loading of vitamin E and carminic acid in Pickering emulsions stabilized by core-mesoporous shell silica nanoparticles. The main difference with previous studies is that they took advantage of the multiple possibilities offered by silica particles for chemical functionalization, conjugating the carminic acid to the particle shell, stabilizing the emulsions, whereas vitamin E remains encapsulated within the oil phase. The use of this type of encapsulation methodology allows better protection of vitamin E against oxidation than conventional emulsions.

Sharkawy et al. [69] explored the use of particles obtained by combining chitosan and collagen for topical applications in the skin, which allows the fabrication of O/W emulsions using olive oil for the oil phase. These emulsions present droplets with an average diameter in the range of 8–16 µm and a high penetration capacity through the stratum corneum. This ensures an efficient deposition of the formulation in deep skin layers. This penetration may be tuned by changing the particle concentration and the contact time of the emulsion with the skin. Further studies on the application of Pickering emulsions for topical delivery were performed by Marto et al. [44]. They fabricated formulations based on Pickering emulsions stabilized by calcium carbonate particles using a Quality by Design approach and found that the optimal formulations were characterized by a shear thinning behavior with a network structure. This favors the spreadability of the formulation on the skin. On the other hand, in vitro cytotoxicity studies evidenced that emulsions stabilized with calcium carbonate particles do not alter cell viability. Moreover, the use of Pickering emulsions facilitate the penetration/permeation of active ingredients toward the inner region of the skin, whereas the particles remain excluded, as demonstrated in the studies by Simovic et al. [70] using an ex vivo porcine skin model.

Table 1 summarizes some of the Pickering emulsions that have been studied in recent years for their cosmetic interest.

**Table 1.** Summary of Pickering emulsions studied for their cosmetic interest, and their potential application.

| Type | Particles | Oil Phase | Application | References |
|------|-----------|-----------|-------------|------------|
| O/W | quinoa starch | Miglyol paraffin sheanut oil | skin care | Marku et al. [63] |
| O/W | PLGA/PSS | medium chain triglyceride | skin care (tocopheryl acetate delivery) | Wei et al. [65] |
| O/W | chitosan/gum arabic | olive oil | skin care (trans-resveratrol delivery) | Sharkawy et al. [66,67] |
| O/W | core-mesoporous shell silica particles | oil mixture (mineral oil and castor oil) | skin care (vitamin E delivery) | Arriagada et al. [68] |
| O/W | chitosan/collagen | olive oil | skin care | Sharkawy et al. [69] |
| O/W | calcium carbonate | caprylic/capric acid triglyceride | skin care | Marto et al. [44] |

## 5. Trends and Challenges toward the Application of Pickering Emulsions in the Cosmetics Industry

Pickering emulsions offer many attractive features, including their stability, easy formation, tunable droplet size, and possibility of chemical modification, which have opened up many opportunities for their exploitation in a broad range of applications [26]. In fact, Pickering emulsions have only recently received a great deal of attention due to their potential use in the formulation of many cosmetics and personal care products, e.g., skin moisturizers, sunscreen lotion, whitening products, anti-aging products, antiperspirants, deodorants, and hair care products [70,71]. This extensive use of Pickering emulsions in the cosmetics industry is the result of consumer demands for more eco-sustainable and safer products and the industry's need to manufacture low-cost and environmentally friendly products. Moreover, the use of particles in cosmetics can contribute to a better spreading and penetration of the formulations within the target cosmetic substrate [54].

The use of Pickering emulsion as encapsulation platforms for different actives in cosmetics has gained interest in recent years due to their ability to offer protection against degradation, increasing the bioavailability of the ingredients and making a controlled delivery possible [72]. This has prompted significant research activity toward the use of stimuli-responsive species for stabilizing Pickering emulsions in the fabrication of cosmetics and personal care products. This may contribute to enhancing the stability and shelf life of the manufactured formulations, providing the basis for a fast and controlled release of active ingredients as a response to external stimuli, e.g., heating, pH or light irradiation [73]. Moreover, a controlled stabilization and destabilization of the emulsions may contribute to the most targeted activity of cosmetics formulations [28].

An unsolved issue related to the exploitation of Pickering emulsions in commercial products is related to the lack of suitable toxicological information about these systems. This is commonly considered a signal of the safety of Pickering emulsions. However, a case-by-case analysis is required to determine the potential toxicity of Pickering emulsions and, hence, their biological sensitivity/irritation effects in humans. A very promising alternative for minimizing the potential adverse effects involves the use of environmentally friendly and biobased stabilizers [28]. Unfortunately, the use of particles in products targeted for human use continues to be a controversial issue due to their potential toxicity and biopersistence [74]. Another very important issue to be solved related to the use of particle-stabilized emulsions is related to the sensorial evaluation of Pickering emulsions upon application on cosmetic substrates. To date, there is information about the sensorial analysis of different cosmetic actives and even about conventional emulsions. Nevertheless, the information related to the sensorial and textural analysis of Pickering emulsions remains very scarce, even though this is of paramount importance for the development of the applications of Pickering emulsions in cosmetics and personal care products [18]. For instance, the rheological and tribological properties of emulsions play a significant role in the texture and sensorial perception of the consumers during the use of cosmetics [75].

It is clear that the exploitation of Pickering emulsions in cosmetics requires extensive work on formulators for reinventing these systems. Nevertheless, to date, there are no commercialized products based on this technology, even though there are different patents dealing with the use of Pickering emulsions in the cosmetics industry. However, several challenges related to the understanding of the most critical physico-chemical factors for developing novel formulations should be solved.

## 6. Conclusions

Pickering emulsions are gaining interest in the cosmetics industry as result of the combination of their enhanced stability in relation to conventional emulsions with many other beneficial properties for final products. For instance, Pickering emulsions can facilitate the vectorization of specific actives in topical applications. Moreover, the use of Pickering emulsions in cosmetics offers several advantages (e.g., dosage, safety, non-toxicity, stability and great absorption for skin) that can contribute to the development of more efficient

products. However, the application of Pickering emulsions in cosmetics needs to overcome several challenges to ensure a satisfactory substitution of molecular emulsifiers, which needs to optimize the production processes of particles and emulsions. This will allow the exploitation of the entire power of Pickering emulsions, which may also present a very important role in the improvement of the sensorial perception associated with the use of emulsions in cosmetics. Despite the potential of Pickering emulsions, their commercialization requires additional in vivo tests because, to date, most of the data related to the cosmetic effectiveness of this type of emulsion have been obtained by in vitro tests, which limits the true validation of their potential application. This review has tried to provide a general overview of the most relevant aspects associated with the use of Pickering emulsions in cosmetics. It is clear that a real implementation of Pickering emulsions in cosmetics can open up very interesting avenues for the optimization of novel formulations. However, it will be necessary to extend the research activity in this direction to ensure a true transference of the fundamental research to an effective cosmetic application.

**Author Contributions:** E.G., F.O. and R.G.R. have contributed equally to the work. All authors have read and agreed to the published version of the manuscript.

**Funding:** This work was funded by MICINN under grant PID2019-106557GB-C21, and by E.U. on the framework of the European Innovative Training Network—Marie Sklodowska-Curie Action NanoPaInt (grant agreement 955612).

**Institutional Review Board Statement:** Not applicable.

**Informed Consent Statement:** Not applicable.

**Data Availability Statement:** Not applicable.

**Conflicts of Interest:** The authors declare no conflict of interest. The funders had no role in the design of the study; in the collection, analyses, or interpretation of data; in the writing of the manuscript, or in the decision to publish the results.

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
