# Peer review of "Pickering Emulsions: A Novel Tool for Cosmetic Formulators"

_cosmetics, doi:10.3390/cosmetics9040068_

Round 1

Reviewer 1 Report

The topic in this review is concentrated on the Pickering emulsion as a new tool for the cosmetic formulation. Authors overview the general aspects of Pickering emulsion, particles in cosmetics and Pickering emulsions in cosmetics. This work is interesting. It can be recommended to be published in the Journal after minor revisions are made as follows.

1. This paper discusses the topic on the Pickering emulsions as a new tool for the cosmetic formulation. Authors should give the more viewpoints to state the importance on the topic.

2. In the section of Pickering emulsions in cosmetics, it is better to list some sub-sections to illustrate the applications of Pickering emulsions in cosmetics.

3. It should give some introduction on the advantage, the problems and the future for the application of the Pickering emulsions in cosmetics.

Author Response

The topic in this review is concentrated on the Pickering emulsion as a new tool for the cosmetic formulation. Authors overview the general aspects of Pickering emulsion, particles in cosmetics and Pickering emulsions in cosmetics. This work is interesting. It can be recommended to be published in the Journal after minor revisions are made as follows.

  1. This paper discusses the topic on the Pickering emulsions as a new tool for the cosmetic formulation. Authors should give the more viewpoints to state the importance on the topic.

Following the reviewer suggestion, we have provided further details stating the interest of the topic.

  1. In the section of Pickering emulsions in cosmetics, it is better to list some sub-sections to illustrate the applications of Pickering emulsions in cosmetics.

The suggestion of the reviewer is very interesting. Unfortunately, the number of examples reported in the literature about Pickering emulsions in cosmetics is relatively low, and hence the splitting of the section would not contribute to a good text flow.

  1. It should give some introduction on the advantage, the problems and the future for the application of the Pickering emulsions in cosmetics.

Following the reviewer recommendation, we have added an additional section discussing some of the most important trend and challenges on the use of Pickering emulsions in cosmetics.

We thank to the reviewer for the comments, they were very useful for improving the quality of the manuscript.

Reviewer 2 Report

Reviewer’s comments:

In this review, the authors represented the basic principle of Pickering emulsion formation and its application prospect in cosmetics. The length of the section “Pickering emulsions in cosmetics” was suggested to be increased and the author should provide a more thorough discussion on the application status and development tendency of Pickering emulsions in cosmetics.

Further comments:

1.   In Table 1, the type of the listed Pickering emulsions by the authors are all O/W and their application involve only skin care. Is the summary on the dosage forms and application adequate?

2.  In the conclusion of the review, the authors mentioned the advantages of Pickering emulsions with low cost, but the authors did not compare the costs of solid particles commonly used in Pickering emulsions with conventional molecular stabilizers.

3.  All the figures look blurry in the review, the authors should change it to clearer figures with higher resolution.

Author Response

In this review, the authors represented the basic principle of Pickering emulsion formation and its application prospect in cosmetics. The length of the section “Pickering emulsions in cosmetics” was suggested to be increased and the author should provide a more thorough discussion on the application status and development tendency of Pickering emulsions in cosmetics.

Following the reviewer recommendation, we have added an additional section discussing some of the most important trend and challenges on the use of Pickering emulsions in cosmetics.

Further comments:

  1.  In Table 1, the type of the listed Pickering emulsions by the authors are all O/W and their application involve only skin care. Is the summary on the dosage forms and application adequate?

Unfortunately, the number of examples reported in the literature about Pickering emulsions in cosmetics is relatively low, and on the best of our knowledge most of them correspond to O/W emulsions for skin care applications.

  1.  In the conclusion of the review, the authors mentioned the advantages of Pickering emulsions with low cost, but the authors did not compare the costs of solid particles commonly used in Pickering emulsions with conventional molecular stabilizers.

This point is frequently stated when the substitution of surfactant for particles on the stabilization of emulsions is discusses. Unfortunately, there are not data available about the cost differences. Therefore, we have decided to remove this point from the conclusions

  1.  All the figures look blurry in the review, the authors should change it to clearer figures with higher resolution.

We have modified the figures for other with higher resolution

We thank to the reviewer for the comments, they were very useful for improving the quality of the manuscript.

Reviewer 3 Report

manuscript is nicely written and presented. In my opinion authors can include some literature about the future trends in pickering emulsions and the Challenges to overcome for an industrial application of Pickering emulsions.

Author Response

manuscript is nicely written and presented. In my opinion authors can include some literature about the future trends in pickering emulsions and the Challenges to overcome for an industrial application of Pickering emulsions.

Following the reviewer recommendation, we have added an additional section discussing some of the most important trend and challenges on the use of Pickering emulsions in cosmetics.

We thank to the reviewer for the comments, they were very useful for improving the quality of the manuscript.

Reviewer 4 Report

1. The limitations of the most common used emulsifiers should be explained in detail.

2. In section 4, it would be great to divide in subsections, explaining the type of cosmetic emulsion product and the used of pickering agents in each type.

Author Response

  1. The limitations of the most common used emulsifiers should be explained in detail.

We have enriched the text with some additional information about the limitations of common emulsifiers.

  1. In section 4, it would be great to divide in subsections, explaining the type of cosmetic emulsion product and the used of pickering agents in each type.

The suggestion of the reviewer is very interesting. Unfortunately, the number of examples reported in the literature about Pickering emulsions in cosmetics is relatively low, and hence the splitting of the section would not contribute to a good text flow.

We thank to the reviewer for the comments, they were very useful for improving the quality of the manuscript.

Reviewer 5 Report

This article entitled “Pickering emulsions: a novel tool for cosmetic formulators” is interesting, however, there are some weak points that should be addressed.

1.     Figures 1, 2, and 3 are not clear. Please increase the resolution of the figures.

2.     In section 2.1, please state in more detail on the preferable methods used to prepare the Pickering emulsions.

3.     In section 2.2, the authors try to explain the important factors affecting the stability of Pickering emulsions, e.g., contact angle, phase volume ratio, trapping energy related to the size of particles as well as particle shape, roughness and charge, and the environmental conditions. I think one more important factor that should be mentioned is “particle size distribution”.

4.     In section 2.3, I think the most preferable rheology of cosmetic products should be pseudoplastic flow, right? However, when a high concentration of solid particles is used, will the rheology of the products change to an undesirable dilatant flow? Please discuss more about this.

5.     Section 3 “Particles in cosmetics”, the subject title seems not related to Pickering emulsions. I suggest modifying this section into “Particles used in Pickering emulsion”.

Author Response

Reviewer 5:

This article entitled “Pickering emulsions: a novel tool for cosmetic formulators” is interesting, however, there are some weak points that should be addressed.

  1. Figures 1, 2, and 3 are not clear. Please increase the resolution of the figures.

We have changed the figures for improving their quality.

  1. In section 2.1, please state in more detail on the preferable methods used to prepare the Pickering emulsions.

We have extended the discussion about preferable methods used for the preparation of Pickering emulsions.

  1. In section 2.2, the authors try to explain the important factors affecting the stability of Pickering emulsions, e.g., contact angle, phase volume ratio, trapping energy related to the size of particles as well as particle shape, roughness and charge, and the environmental conditions. I think one more important factor that should be mentioned is “particle size distribution”.

We have discussed in the text the point raised by the reviewer.

  1. In section 2.3, I think the most preferable rheology of cosmetic products should be pseudoplastic flow, right? However, when a high concentration of solid particles is used, will the rheology of the products change to an undesirable dilatant flow? Please discuss more about this.

The point raised by the reviewer is very interesting. Unfortunately, there are no enough information of the literature for such claiming, and we prefer to avoid discussions without enough previous support.

  1. Section 3 “Particles in cosmetics”, the subject title seems not related to Pickering emulsions. I suggest modifying this section into “Particles used in Pickering emulsion”.

We have changed the title of the section according to the reviewer suggestion.

We thank to the reviewer for the comments, they were very useful for improving the quality of the manuscript.

Round 2

Reviewer 2 Report

The manuscript can be accepted in present form.

Reviewer 5 Report

The author's responses and revisions are clear and correct.